# Enhancing clinical decision-making: Sysmex UF-5000 as a screening tool for bacterial urinary tract infection in children

Ping Liu[1,2], Chuanwei Ban[1,2], Juan Wang[1,2], Qian Zeng[1,2], Mengmeng Chen[1,2], Ling Wang[1,2]*, Xin Lv[1,2]*

1 Clinical Laboratory, Children's Hospital Affiliated to Shandong University, Jinan, China, 2 Clinical Laboratory, Jinan Children's Hospital, Jinan, China

* etyyjyklvxin@163.com (XL); 457906059@qq.com (LW)

## Abstract

### Background

A rapid screening test for urinary tract infections (UTIs) in children is needed to avoid unnecessary cultures and provide prompt reports to make appropriate clinical decisions. We have evaluated for the first time the performance of the Sysmex UF-5000 flow cytometer as a screening tool for UTIs in children.

### Methods

This study included 4445 pediatric patients, with urinary sediment and urine culture data collected from January 2020 to September 2023. The Sysmex UF-5000 analyzer was utilized to measure urine white blood cell (WBC) and bacteria (BACT), with the findings being compared to the culture results.

### Results

At $\geq 10^4$ colony-forming unit (CFU)/mL, 513 samples were culture-positive (400 samples presented $10^4$–$10^5$ CFU/mL, and 113 demonstrated $\geq 10^5$ CFU/mL bacterial growth). Optimal indicators for positive cultures were BACT counts of 92.2/μL (AUC: 0.944) and WBC counts of 40.8/μL (AUC:0.863). False negative rate were 0.9% when using a 7.8 bacteria/μL cut-off and avoiding unnecessary cultures in 28.1%. The UF-5000 has a higher consistency rate for Gram-negative (GN) bacteria (90.3%) than Gram-positive (GP) bacteria (86.8%). For samples with $10^5$ CFU/mL, UF-5000's Bacteria -Information flags showed superior concordance for samples with $10^4$–$10^5$ CFU/mL bacteria.

### Conclusions

Screening pediatric urine cultures with the UF-5000 showed potential application value in identifying negative cultures and significant bacterial growth, although performance may vary depending on the study population. Furthermore, detecting Gram typing aids in guiding early clinical empirical medication, particularly for UTIs caused by GN bacteria.

**Data Availability Statement:** Our research was based on data extracted from medical records, which contain sensitive patient information. Due to

ethical restrictions imposed by the Children's Hospital Affiliated to Shandong University Ethics Committee, the data cannot be shared publicly. For data access requests, interested parties can reach out to the Ethics Committee at the Children's Hospital Affiliated to Shandong University (contact via phone +0086531596016082). Due to the possibility that the official email may automatically block overseas emails, we have left the Ethics Committee office phone number to process relevant data requests as soon as possible.

**Funding:** The author(s) received no specific funding for this work.

**Competing interests:** The authors have declared that no competing interests exist.

**Abbreviations:** AUC, area under the curve; BACT, bacterial; CFU, colony-forming unit; GN, Gram-negative; GP, Gram-positive; NPV, negative predictive value; PPV, positive predictive value; ROC, Receiver Operating Characteristic; SE, Sensitivity; SP, specificity; UTI, urinary tract infection; WBC, white blood cell.

## Introduction

Urinary tract infections (UTIs) are among the most prevalent infectious diseases during childhood [1, 2]. The clinical signs and symptoms of UTIs in children are complex and diverse, and urinary symptoms in infants and young children are not obvious. If not treated promptly, it can readily progress to nephritis and even renal failure [3]. Urine culture remains the gold standard technique for the species identification and diagnosis of UTIs [4]. However, this method usually requires 2 to 3 days for final reporting, which does not meet the requirements for rapid clinical diagnosis. Moreover, clinical microbiology laboratories commonly receive culture-negative specimens, which increases the workload [5, 6]. In addition, it is more difficult for children to retain midstream urine than adults, and improper collection is more likely to lead to contamination and hence false positive results. Therefore, a rapid screening test for UTIs in children is needed to avoid unnecessary cultures and provide prompt reports to make appropriate clinical decisions.

Previously, due to the unavailability of timely laboratory results, clinicians would usually administer antibiotic treatments empirically without supportive laboratory evidence, which led to the overuse of antibiotics and increased the risk of drug resistance [7]. Therefore, rapid and accurate pathogen identification is crucial for effective anti-infection treatments.

Urine dipsticks are frequently utilized as a fast initial screening test, despite having limited sensitivity(SE) [8, 9]. Flow cytometry, a technique capable of distinguishing and quantifying various particles (including bacteria) in urine, offers an alternative approach for rapidly screening UTIs. The Sysmex UF-5000 (Sysmex, Kobe, Japan), a third-generation fully automated urine flow cytometer, is a notable advancement over its predecessor, the UF-1000i (Sysmex). The most significant enhancement is the addition of a new parameter for bacterial information (BACT-Info) in the UF-5000, which distinguishes bacterial Gram-staining subtypes based on their different uptake of pigments through cell wall structure.Gram-positive (GP) bacteria have a thicker peptidoglycan layer, leading to increased forward scattered light intensity (FSC) and less pigment penetration, resulting in lower lateral fluorescence intensity (FL), and tend to be located in regions with larger angles on scatter plots. Conversely, Gram-negative (GN) bacteria have a thinner peptidoglycan layer, producing weaker FSC and allowing more pigment penetration, resulting in higher FL, and are typically found in flatter areas with smaller angles.

In recent years, some studies have shown that among the 17 parameters available for urine analysis on the Sysmex UF-5000, white blood cell (WBC) counts, BACT counts and BACT-Info are the most useful for UTI diagnosis [6, 10–13]. However, the results exhibit high heterogeneity due to differences in patient populations, specimen types, and selected thresholds for significant counts in culture. Most previous studies have focused on adult samples. Consequently, this study explored the screening performance of the novel UF-5000 fluorescence flow cytometer, specifically in pediatric UTIs, and its ability to differentiate Gram typing of bacteria in UTI samples using new parameters.

## Materials and methods

### Patients and specimen collection

A total of 4445 pediatric patients with urinary sediment and urine culture information were retrospectively analyzed at Children's Hospital Affiliated to Shandong University between January 2020 and September 2023. Among the patients, 2424 (54.5%) were male and 2021(45.5%) were female. Hospitalized patients accounted for 4183 (94.1%), and 262 (5.9%) were outpatients. The median age was 2 years (range, 0–17 years). The specimen collection was completed

clinically, and clean midstream urine samples were collected from toilet-trained children. In contrast, young infants who had not yet been toilet-trained had their samples collected in a sterile collection bag after careful washing of the genitals, with urine flow being monitored. All specimens were collected in two ordinary sterile test tubes for i) urine culture and ii) flow cytometry analyses. Samples were transported to the laboratory within 2 h of collection.

## Urine culture and identification of microbes

Inoculation involved using a calibrated loop to apply 10μL of well-mixed urine specimen onto blood agar and chocolate agar plates for quantitative cultivation. All plates were incubated for 18–24 h at 37˚C in 5% $CO_2$. If there was bacterial growth, the colonies were counted. If there was no growth, cultivation continued for 48h.BACT counts $\geq 10^4$ CFU/mL were considered culture positive, BACT counts $< 10^4$CFU/mL raised suspicion of contamination [14]. The urine was classified as contaminated if three or more colonies without a predominant organism were observed, representing the diagnostic standard for pediatric patients in the author's institution. Bacteria were identified using a Microscan WalkAway 96 Plus automatic bacterial identification instrument (SIEMENS, Germany).

## Sysmex UF-5000 analysis

This instrument uses flow cytometry to detect sediment in urine. Based on nucleic acid fluorescence staining principles and semiconductor laser flow cytometry, it can provide 14 conventional quantitative and qualitative detection parameters and 3 clinical parameters, such as detailed information on red blood cells, conductivity information, and BACT-Info [15]. As with the previous analyzers in the UF-Series (UF-100 and UF-1000i), the UF-5000 is very useful in screening and identifying UTIs. Moreover, BACT-Info displays as "Gram-positive?" [GP], "Gram-negative?" [GN], "Gram-pos/neg?" [GP/GN] or "Unclassified." Thus, in addition to providing the same information as traditional Gram staining, the UF-5000 offers a faster and simpler alternative.

## Statistical analysis

The GP and GN discrimination results from the UF-5000 were compared with urine culture results. Continuous variables were expressed as medians and ranges. The Mann-Whitney U test was used to compare continuous variables between the two studies. Receiver operating characteristic (ROC) curve analyses were performed, and the area under the curve (AUC) was used to assess the diagnostic accuracy of BACT and WBC counts for UTIs. SE, specificity (SP), positive predictive value (PPV), and negative predictive value (NPV) were calculated at different BACT cut-offs using urine culture as a reference. The largest Youden index (SE+ SP-1), was utilized to determine the optimal cut-off points for distinguishing positive and negative samples. Statistical analysis was conducted using SPSS software (version 23.0; SPSS, Chicago, IL), and all figures were created with GraphPad Prism (GraphPad Software, San Diego, USA). $P < 0.05$ was considered statistically significant.

## Ethics statements

According to the current version of the Declaration of Helsinki, this study received approval from the Ethical Committee of Children's Hospital Affiliated to Shandong University (SDFE-IRB/T-2023095) and commenced data on December 29, 2023.The authors were granted access to identifiable participant information during the data collection process.

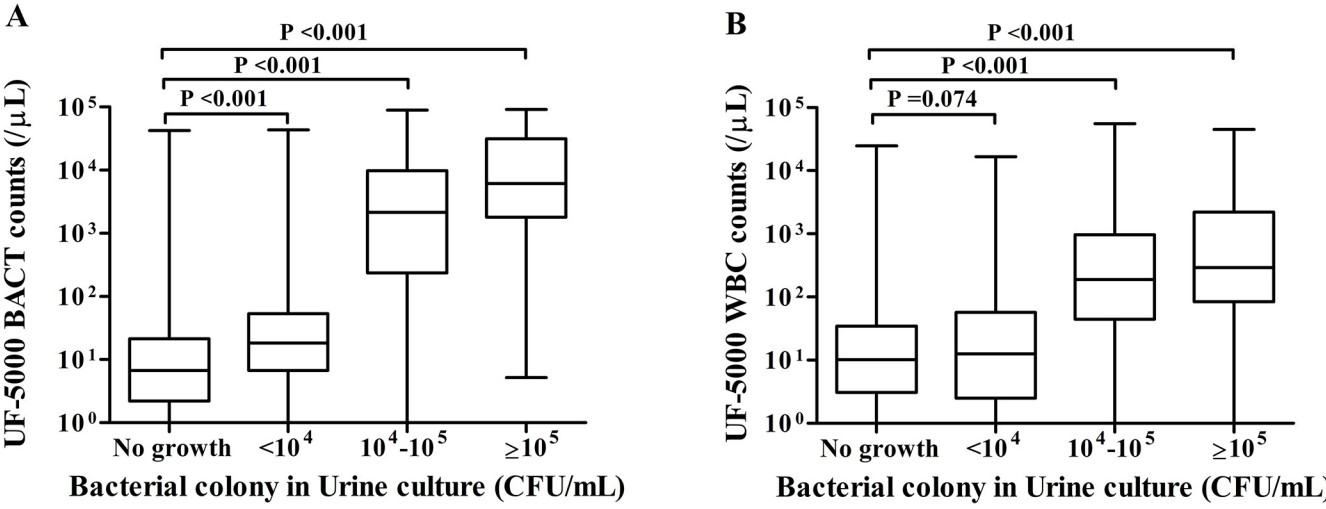

**Fig 1.** The BACT(A)and WBC(B)counts detected by the UF-5000 under different bacterial colony counts in urine cultures among total samples excluding fungi.

## Results

### UF-5000 BACT and WBC counts

Of 4445 urine cultures, 3443 (77.5%) displayed no growth, 435 (9.8%) exhibited growth of $<10^4$ CFU/mL of bacteria, 400 (8.9%) showed growth between $10^4$–$10^5$ CFU/mL, and 113 (2.5%) presented growth $\geq 10^5$ CFU/mL of bacteria. Fungal growth was detected in 54 (1.2%) samples.

The distribution of BACT and WBC counts with the UF-5000 under different colony counts in urine cultures is shown in Fig 1. The median of bacteria was 8.8/μL (range: 0–42566.2/μL) for the group with no growth, 18.8/μL (range: 0–43647.0/μL) for the $<10^4$ CFU/mL group; 2166.2/μL (range: 1.0–89503.1/μL) for the $10^4$–$10^5$ CFU/mL group; and 6183.3/μL (range: 5.2–90908.6/μL) for the $\geq 10^5$ CFU/mL group. BACT counts in all groups were significantly higher than in samples without growth (P < 0.001). Similarly, when classified by colony counts in urine culture, the median of WBC was 10.2/μL (range: 0–24851.3/μL) for the no-growth group, 12.7/μL (range: 0–16674.3/μL) for the $<10^4$ CFU/mL group, 188.4/μL (range: 0.4–55528.2/μL) for the $10^4$–$10^5$ CFU/mL group, and 292.3/μL (range: 0.5–44706.6/μL) for the $\geq 10^5$ CFU/mL group. In samples with bacterial growth ($10^4$–$10^5$ and $\geq 10^5$ CFU/mL), the WBC counts were significantly higher than those in samples without growth (P < 0.001).

### Microbial species

We identified 513 culture-positive samples. Among these, 471 samples (91.8%) were identified as single microorganisms (303 samples [59.1%] with GN bacteria and 168 samples [32.7%] with GP bacteria). In 42 samples (8.2%), two microorganisms were isolated, 17 cultures presented mixed GN and GP bacterial growth, and 25 showed two GN bacteria. The microorganisms isolated reflected the usual rate of uropathogens in our laboratory. The most frequently reported bacteria were GNs, constituting 60.6% of all species isolated, with *Enterobacterales* being the majority. GP pathogenic bacteria are mainly attributed to *Enterococcus faecium* and *Enterococcus faecalis*. Yeast growth was observed in 54 samples, of which 27 were *Candida albicans* and 15 were *Candida tropicalis* (Table 1).

**Table 1. Microorganisms identified in the 567 urine culture samples.**

| Microorganism | No. of samples (n) | Proportion of samples (%) |
|---|---|---|
| **Gram positives** | **186** | **30.5** |
| *Enterococcus faecium* | 91 | 14.9 |
| *Enterococcus faecalis* | 76 | 12.5 |
| *Streptococcus agalactiae(Group B)* | 4 | 0.7 |
| *Staphylococcus aureus* | 2 | 0.3 |
| Other Gram positives[a] | 13 | 2.1 |
| **Gram negatives** | **369** | **60.6** |
| *Escherichia coli* | 174 | 28.7 |
| *Klebsiella spp.* | 69 | 11.4 |
| *Pseudomonas aeruginosa* | 35 | 5.7 |
| *Proteus mirabilis* | 30 | 4.9 |
| *Enterobacter cloacae* | 19 | 3.1 |
| *Citrobacter freundii complex* | 8 | 1.3 |
| *Acinetobacter baumannii* | 7 | 1.1 |
| Other Gram negatives[b] | 27 | 4.4 |
| **Fungus** | **54** | **8.9** |
| *Candida albicans* | 27 | 4.4 |
| *Candida tropicalis* | 15 | 2.5 |
| *Candida glabrata* | 8 | 1.3 |
| Other Fungi[c] | 4 | 0.7 |

[a] Other Gram positives included *Enterococcus gallinarum (2)*, *Enterococcus avium (2)*, *Streptococcus gallolyticus (2)*, *Corynebacterium striatum (2)*, *Streptococcus anginosus (2)*, *Staphylococcus epidermidis (2)*, *Staphylococcus saprophyticua (1)*.

[b] Other Gram negatives bacteria included *Klebsiella oxytoca (5)*, *Morganella morganii (5)*, *Stenotrophomonas maltophilia (5)*, *Citrobacter amalonaticus (3)*

Citrobacter koseri (2), Proteus vulgaris (2), Burkholderia multivorans (1), Ralstonia mannitolilytica (1), Providencia rettgeri (1), Alcaligenes xylosoxidans (1), Serratia marcescens (1).

[c] Other Fungi included *Candida parapsilosis (3)* and *Candida krusei (1)*.

## Diagnostic performance of UF-5000

Excluding samples from the fungus culture and the $< 10^4$ CFU/mL group, a ROC curve is presented in Fig 2 for Sysmex UF-5000 BACT and WBC counts, with urine cultures as the reference. The AUC for UF-5000 in BACT counts was 0.944 (95% CI: 0.933 to 0.956) higher than the AUC for WBC counts;0.863 (95% CI: 0.846 to 0.881).

We evaluated the SE and SP of the UF-5000 BACT counts. The largest Youden index showed that the best cut-off value for UF-5000 BACT counts on the ROC curve was 92.2/μL, resulting in a SE of 86.0% (95% CI: 0.827 to 0.889), SP of 91.4% (95% CI: 0.905 to 0.924), and approximately 79.6% as unnecessary urine cultures, with 14.6% false-negative results. The results are summarized in Table 2. To minimize false negatives, we imposed higher sensitivities (90%, 95%, and 99%) as a condition, yielding cut-off values of 47.5 bacteria/μL, 13.5 bacteria/μL, and 7.8 bacteria/μL, with corresponding false negative rates of 9.7%, 4.5%, and 0.9%, respectively. As shown in Table 2, when the maximum SE was reached, the bacterial count indicated that UF-5000 could eliminate 28.1% of unnecessary cultures.

At the 7.8 bacteria/μL cut-off, five false negatives were identified: four samples with growth of GP cocci and one sample with mixed growth of two GN bacteria. At the 13.5 bacteria/μL cut-off, 23 false negatives were observed: 15 GP cocci and 8 GN bacteria.

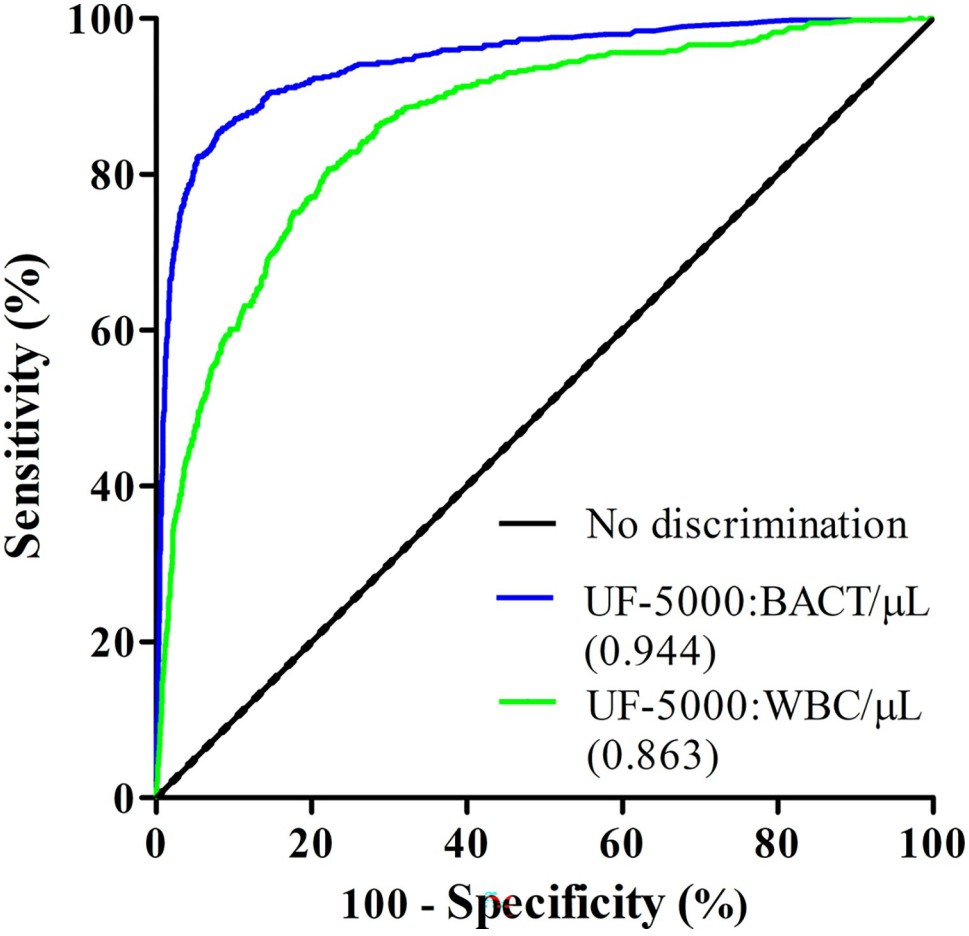

**Fig 2. ROC curve of BACT and WBC counts by UF-5000 flow cytometer in 3956 samples.** (with positive cut-off value at $1 \times 10^4$ CFU/mL).

WBC count also plays a vital role in detecting UTIs and assessing treatment effectiveness because it provides an estimate of asymptomatic bacteriuria or contamination due to infection [6]. We determined a general cut-off point of 40.8 WBCs/μL, providing the most significant

**Table 2. Performance of UF-5000 with different cut-off values for BACT counts in 3956 samples.**

| Performance | Cut-off BACT Count (/μL) | | | |
|---|---|---|---|---|
| | **92.2** | **47.5** | **13.5** | **7.8** |
| Sensitivity % | 86.0 | 90.0 | 95.0 | 99.0 |
| Specificity % | 91.4 | 85.6 | 66.4 | 32.3 |
| Negative Predictive value (NPV) % | 97.7 | 98.3 | 99.0 | 99.6 |
| Positive Predictive value (PPV) % | 59.8 | 48.3 | 29.8 | 17.9 |
| False negative rate % | 14.6 | 9.7 | 4.5 | 0.9 |
| Unnecessary cultures % | 79.6 | 74.5 | 57.8 | 28.1 |
| True positives (n) | 438 | 463 | 490 | 508 |
| False positives (n) | 294 | 495 | 1156 | 2332 |
| True negatives (n) | 3149 | 2948 | 2287 | 1111 |
| False negatives (n) | 75 | 50 | 23 | 5 |

**Table 3. Comparison of the UF-5000 BACT-Info flags and urine culture results.**

| BACT-Info flag by UF-5000 | Urine Culture | | | |
|---|---|---|---|---|
| | Gram-negative bacterial growth (n,%) | Gram-positive bacterial growth (n,%) | Gram-negative and positive mixed growth (n,%) | No growth (n,%) |
| Total specimens | | | | |
| Total | 328 | 168 | 17 | 3443 |
| Gram-negative? | 202 (61.6) | 6 (3.6) | 4 (23.5) | 33 (0.9) |
| Gram-positive? | 5 (1.5) | 64 (38.1) | 3 (17.6) | 20 (0.6) |
| Gram-pos/neg? | 41 (12.5) | 13 (7.7) | 2 (11.8) | 20 (0.6) |
| Unclassified | 80 (24.4) | 85 (50.6) | 8 (47.1) | 3370 (97.9) |
| $\geq 10^5$ CFU/mL | | | | |
| Total | 81 | 30 | 2 | NA |
| Gram-negative? | 60 (74.1) | 4 (13.3) | 0 | NA |
| Gram-positive? | 0 | 16 (53.4) | 0 | NA |
| Gram-pos/neg? | 10 (12.3) | 3 (10.0) | 1 (50.0) | NA |
| Unclassified | 11 (13.6) | 7 (23.3) | 1 (50.0) | NA |
| $10^4-10^5$ CFU/mL | | | | |
| Total | 247 | 138 | 15 | NA |
| Gram-negative? | 142 (57.5) | 2 (1.4) | 4 (26.7) | NA |
| Gram-positive? | 5 (2.0) | 48 (34.8) | 3 (20.0) | NA |
| Gram-pos/neg? | 31 (12.6) | 10 (7.2) | 1 (6.7) | NA |
| Unclassified | 69 (27.9) | 78 (56.6) | 7 (46.6) | NA |

NA, not assessed.

area under the ROC curve and the best Youden index. The SE, SP, PPV, and NPV were 80.7% (95% CI: 0.770 to 0.840), 77.8% (95% CI: 0.761 to 0.792), 96.5%, and 35.2%, respectively.

## Performance evaluation of UF-5000 in distinguishing GP and GN bacteria

The discrimination results of UF-5000 BACT-Info flags versus urine culture are shown in Table 3. Samples with less significant growth or fungus growth were excluded. Of the total specimens, based on the gold standard of midstream urine cultures, the total consistency rate of UF-5000 for distinguishing UTIs is 92.0% (3638/3956), the consistency rate for determining GN in urine is 90.3% (3572/3956), the SE is 61.6% (202/328), the consistency rate for GP is 86.8% (3434/3956), the SE is 38.1% (64/168), the consistency rate for GP/GN is 85.2% (3372/3956), and the SE is 11.8% (2/17). For samples with $\geq 10^5$ CFU/mL bacteria, "Gram-negative?" UF-5000 BACT-Info flags showed a better concordance of 74.1% (60/81) compared to $10^4-10^5$ CFU/mL bacteria, with 57.5% (142/257) concordance. Similarly, 53.4% (16/30) of urine cultures with growth $\geq 10^5$ CFU/mL of GP bacteria showed higher concordance than urine cultures with $10^4-10^5$ CFU/mL 34.8% (48/138) (Table 3). Over a third of samples with bacterial growth at $< 10^5$ CFU/mL exhibited "unclassified" flags.

## Discussion

Urine culture remains the gold standard for diagnosing UTIs. However, the testing cycle is long and unsuitable for early diagnosis. It necessitates establishing a rapid and accurate primary screening method for UTIs in children. This is the first study to compare the application value of the new-generation Sysmex UF-5000 flow cytometry in children with bacterial UTIs

with urine cultures. It was also the most significant number of analyzed urine samples among the UF-5000 studies, with 4445.

In this study, positive urine culture results were 13.0% (513/3956). This result is lower than that reported in previous studies involving adults, researchers report 20% -40% of positive cultures in most studies [6, 12, 13, 16], but 10.4% in one study of children [17], aligning with our laboratory results. The reasons for this variance may be due to the non-specific clinical manifestations of children, making the diagnosis of UTIs in the pediatric population challenging. The use of antibiotics in children before specimen collection may lead to false negatives in traditional culture results [14]. In addition, children are unlikely to retain urine for an extended period, and bacteria may not have enough time to reproduce in the bladder if urine remains for less than 4 hours.GN bacteria predominated in positive cultures, accounting for 60.6%, of which most were *Escherichia coli*. Therefore, as described in previous studies [2, 3, 18–20], *Enterobacterales* especially *E. coli*, cause UTIs in most children. *Enterococcus* is the second most common pathogen in our study, accounting for 27.4%, which may be attributed to the predominance of hospitalized patients, aligning with literature that typically describes *Enterococcus* as a pathogen associated with hospitalization [8, 21].

We compared the BACT and WBC counts detected by the UF-5000 under different bacterial colony counts in urine. The findings suggest a strong correlation between UF-5000 BACT, WBC counts, and standard urine culture results. In other words, those with higher colony counts also have higher BACT and WBC counts. The BACT and WBC counts in the samples with bacterial growth $10^4$–$10^5$ and $\geq 10^5$ CFU/mL were significantly higher than those with no growth (P < 0.001). Although there were statistically significant differences in BACT counts in the $< 10^4$ CFU/mL group, no significant increase in WBC counts was observed compared to the no-growth group, a low risk of UTI and a high risk of colonization or contamination are the most likely explanations.

Previous studies have shown high variability in proposed cut-off points for the Sysmex auto analyzer, which is probably related to each center's different UTI diagnosis criteria, disease prevalence rates, and study samples sources.

For BACT and WBC counts in urinary samples, a recent study showed that UF-5000 exhibited acceptable SEs (96.1% and 91%) and lower SPs (32.2% and 28%) at 100 bacteria/μL and 21 WBCs/μL, respectively [16]. In another study, a BACT count at a cut-off of 135/μL, the SE and SP were 92.1% and 85.4%. Furthermore, there was a WBC cut-off of 23 /μL, and the SE and SP were 73.5% and 84.1% [12]. Our optimal cut-off points for positive urine culture results, following the criterion of the largest Youden index, would be 92.2/μL for BACT (SE: 86.0%, SP: 91.4%) and 40.8/μL for WBC (SE: 80.7%, SP: 77.8%). There is no standardized cut-off point for urine screening, and each laboratory should set its threshold dependent on the patient's history and the specific criteria for urine culture.

ROC curve analysis showed that the AUC of the WBC counts was 0.863 and that of the BACT was 0.944, indicating that both the WBC and BACT counts have diagnostic values for UTIs. However, in our evaluation of UF-5000, we found a higher AUC for BACT than for WBC, as shown in previous studies [13, 22, 23]. In other words, the WBC count was less effective than the BACT count for screening UTIs. The results of our analysis determined data on the SE and SP of different UF-5000 BACT count cutoffs. With the best cut-off value, the percentage of unnecessary urine cultures was approximately 79.6%; however, the false negative rate was 14.6%. Given this, our laboratory selects lower cut-off values to achieve a higher sensitivity to screen for UTIs. With 7.8 bacteria/μL set as the threshold, the false negative rate is only 0.9%; there were 1111 samples with < 7.8 bacteria/μL and culture-negative. When BACT counts were done with the UF-5000 first, 1111 samples (28.1% of the total specimens) would not require culture if there were no more than 7.8 bacteria/μL in the urine sample. This would reduce the costs of urine culture and workload.

Additionally, there is substantial variability between the various studies published in this field. Alenkaer et al. reduced unnecessary cultures by 30% with 99% sensitivity [11]. However, De Rosa et al. reached 55% [6]. In addition, Toledo et al. reduced unnecessary cultivation by 40% at a sensitivity of 95% [16], but these data are all from studies on adults. Our laboratory achieved a 57.8% reduction in unnecessary cultivation at a sensitivity of 95%.

It should be noted that most of our false negative cultures were GP cocci, accounting for 80% (4/5) and 65.2% (15/23), respectively. This phenomenon may be related to the tendency of GP cocci to form bacterial aggregates, which interfere with cytometer readings. Toledo et al. [16] and Manoni et al. [24] reported that GP cocci were obtained from over half of the urine samples that were false-negative. This finding is consistent with our conclusions.

Based on previous results, the UF-5000 also provides information on bacterial Gram staining. We evaluated the performance of the UF-5000 in distinguishing GP and GN bacteria. Considering all the samples collectively, our study showed that the total consistency rate of UF-5000 for determining UTIs is 92.0%, indicating that the culture results and bacterial detection by the UF-5000 are highly consistent. Furthermore, GN and GP had 90.3% and 86.8% coincidence rates, respectively, and 61.6% and 38.1% sensitivity, respectively; similar results were reported by Kim et al. [10]. The consistency rate for GP/GN is 85.2%, and the sensitivity is 11.8%, compared to individual bacterial infections, UF-5000 is less effective at subtyping mixed UTI infections. According to our research, the UF-5000 is more sensitive to GP and GN bacteria in UTIs with $10^5$ CFU/mL bacterial growth than with $10^4$–$10^5$ CFU/mL bacteria, indicating that samples with lower counts produced inaccurate results, reported as "unclassified."

It's worth noting that for samples with significant growth, the SE of the UF-5000 bacterial channel to detect GP bacteria was also low ($< 60.0$%). This contrasts with previous studies [10, 12], which have indicated a higher rate of missed diagnosis in GP screening for the UF-5000. The results showed that the detection ability of UF-5000 was better for GN bacteria than GP bacteria, given that GN bacteria are the primary causative pathogens of UTIs. The UF-5000 has good detection ability for GN bacteria and can guide clinicians to empirically choose antibiotics for UTI treatment, thereby improving the optimal treatment timing.

In summary, the UF-5000 demonstrates clinical utility in screening urine cultures for significant bacterial growth. While the potential use of its generated BACT counts in excluding UTIs and reducing unnecessary urine cultures, its effectiveness may depend on the study population. We recommend a BACT count cut-off of 7.8/μL for screening out patients with negative cultures. In contrast, before the results of urine cultures, clinicians may use BACT counts and classifications as supplementary information in deciding whether to treat with antibiotics in patients with urological issues.

This study had some limitations. It was conducted at only one center, therefore, multicenter studies involving larger sample sizes are necessary to validate our results. A substantial number of samples showed $\leq 10^5$ CFU/mL bacterial growth, and due to the random selection of these samples, it cannot be ruled out that most patients have been treated with antibiotics before their hospital visit. Therefore, comprehensive patient history and clinical assessment are crucial. Future research, particularly in children, is needed to optimize cutoff values and develop appropriate algorithms for effectively applying the UF-5000 in clinical practice.

## Supporting information

**S1 Fig. Flowchart of the participants included in the study.**
(TIF)

## Author Contributions

**Conceptualization:** Ping Liu, Ling Wang, Xin Lv.

**Data curation:** Ping Liu, Chuanwei Ban, Juan Wang.

**Formal analysis:** Ping Liu, Qian Zeng.

**Investigation:** Chuanwei Ban, Juan Wang, Mengmeng Chen.

**Methodology:** Ping Liu, Qian Zeng, Ling Wang, Xin Lv.

**Project administration:** Mengmeng Chen.

**Supervision:** Ling Wang, Xin Lv.

**Validation:** Chuanwei Ban, Juan Wang, Qian Zeng.

**Writing – original draft:** Ping Liu.

**Writing – review & editing:** Ling Wang, Xin Lv.

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
