## [Decision Letter · Decision Letter 0]

20 Mar 2024

PONE-D-24-04363Enhancing clinical decision-making: Sysmex UF-5000 as a screening tool for urinary tract infections in childrenPLOS ONE

Dear Dr. Lv,

Thank you for submitting your manuscript to PLOS ONE. After careful consideration, we feel that it has merit but does not fully meet PLOS ONE’s publication criteria as it currently stands. Therefore, we invite you to submit a revised version of the manuscript that addresses the points raised during the review process.

We look forward to receiving your revised manuscript.

Kind regards,

Eleonora Nicolai, PhD

Academic Editor

PLOS ONE

Journal Requirements:

3. For studies involving third-party data, we encourage authors to share any data specific to their analyses that they can legally distribute. PLOS recognizes, however, that authors may be using third-party data they do not have the rights to share. When third-party data cannot be publicly shared, authors must provide all information necessary for interested researchers to apply to gain access to the data. (https://journals.plos.org/plosone/s/data-availability#loc-acceptable-data-access-restrictions) 

Additional Editor Comments:

Dear Authors,

reviewers have analyzed your paper and expressed several point to be addressed. Please revise your manuscript answering to the questions raised by reviewers.

Kind regards

Reviewers' comments:

Reviewer's Responses to Questions

**Comments to the Author**

1. Is the manuscript technically sound, and do the data support the conclusions?

Reviewer #1: Yes

Reviewer #2: Yes

Reviewer #3: Yes

Reviewer #4: Yes

Reviewer #5: Partly

2. Has the statistical analysis been performed appropriately and rigorously? 

Reviewer #1: Yes

Reviewer #2: No

Reviewer #3: Yes

Reviewer #4: No

Reviewer #5: Yes

3. Have the authors made all data underlying the findings in their manuscript fully available?

Reviewer #1: Yes

Reviewer #2: Yes

Reviewer #3: Yes

Reviewer #4: Yes

Reviewer #5: Yes

4. Is the manuscript presented in an intelligible fashion and written in standard English?

Reviewer #1: Yes

Reviewer #2: Yes

Reviewer #3: Yes

Reviewer #4: No

Reviewer #5: Yes

5. Review Comments to the Author

Reviewer #1: This is a very interesting study testing the Sysmex UF-5000 as a screening tool for UTI in children.

Zhe authors included 4445 patient informations and compared the BACT and WBC counts received by the UF-5000 with the measured data in the laboratories. The bacterial counts in the laboratory was classified as „no growth“, <10*4 CFU/ml, 10*4-10*5/ml, and >10*5/ml. The authors calculated for each parameter the usual statistical parameters, like sensitivity and specificity, negative and positive predictive values etc. Depending on the Cut-off BACT they could calculate these statistical parameters with the results, how many urine cultures would have been most likely unnecessary. Because they also could differentiate with high probability between most likely GN and GP cultures, these results would also be important if empiric antibiotic therapy was to be chosen. The authors compared their results with other studies as well, however, most of them were performed in adults. They also discussed well the limitations of their own study.

Overall, the study was well performed and presented. I only have (minor) comments/questions. Unfortunately the lines of the manuscript are not numbered.

1. Concludions: I like that the authors write Gram usually with capital letter, because it is the name of a person. In Conclusion it should also be written with capital letter.

2. Urine culture and identification of microbes.“If there was sterile growth“. Although the authors used 10 µl for urine culture, also in case of no growth, one cannot say ir is sterile. Therefore, better: „If there was no growth“.

Table 2. Typing error: Tocal? Maybe should read Total. It was used 3 times.

Otherwise I have no further comments.

Reviewer #2: Urinalysis is a very important issue in most laboratories worldwide. In the last decades automated urine particle analyzers have reduced microscopic examinations and unnecessary cultures. In their study, Liu P et al. investigated the performance of the UF-5000 fluorescence flow cytometer especially in pediatric urinary tract infections.

This report has some merit because of the pediatric study population. The manuscript is well written, however, before publication in this journal some minor points should be considered:

• Abstract (methods): “We analyzed 4445 patient information of children….” This sentence should be rewritten in a better and more informative way.

• Ethics statement: “And began accessing data on December 29, 2023. The author had access to information that could identify individual participants during data collection…” This part should be rewritten in a better English.

• Results/Microbial species: “C. tropical” should be spelled out to “Candida tropicalis”

• Table 1: “Proteus mirabili” should read “Proteus mirabilis”

• Results/Diagnostic performance of UF-5000: “Excluding samples from the fungus culture and the <104 CFU/mL group, an ROC curve…” should read “…, a ROC curve…”

• The “limitations” should be mentioned before the conclusion.

Reviewer #3: The manuscript “Enhancing clinical decision-making: Sysmex, UF-500 as a screening tool for urinary tract infections in children” is interesting and confirms/validates in a large cohort, previous findings on the utility of flow cytometry for detection of UTI. Several studies have evaluated the performance of the UF-5000 in different kinds of patient cohorts but as this study is the first on a large cohort of children, it brings new knowledge to front. The paper needs some revision before publishing.

DETAILED COMMENTS

Q1: Abstract, Results. The author mentions that unnecessary culture could be avoided in 28.1%. Are there a reason for presented the numbers with 3 digits? Why not say 28%. I hardly think the estimates is that precise?

Q2: Abstract, Conclusion. Cultures screened with UF-5000 showed superior results in identifying…superior in comparison to what? I don’t think you mean compared to urine culture? Or do you mean compared to previous flow cytometry models like UF-1000 or compared to Urine stricks? The UF-5000 may be faster than culture but not better. It may also be better that other fast screening methods.

Q3: Introduction/discussion. The authors mention that rapid screening test for UTI is needed for children. But there is nothing mentioned about urine dipsticks that are ruinously used in many settings for rapid screening, including in children >2 years of age. Is that because the dipsticks are not used at this children’s’ hospital? It may be worth mentioning that there are a lot of rapid screen test on the marked, but they do show inadequate performance compared to culture.

Q4: Urine Culture: Inoculation involved using a calibrated loop to apply 10 µL culture. It is common to use a 1-μL inoculating loop, so can you elaborate on the choice of the 10 µL loop? It may be a good point to include in your discussion as many of the studies you compare uses 1 µL loop cultures and therefore you may not compare to the same culture standard.

Q4: Urine culture. BACT counts≥104CFU/mL should be changed to BACT counts≥104 CFU/mL. Do the sentence BACT counts ≥104CFU/mL (when symptomatic) where considered positive, mean that samples with ≥104CFU/mL where excluded if the patient did not have any symptoms? I don’t think you can exclude culture positive samples from the dataset, at least not without mentioning this a an exclusion criteria. Also how where ≥103CFU/mL classified? As negative? You may stick to the recommended cut-off urine cultures in children at ≥104 CFU/mL.: 1999. Practice parameter: the diagnosis, treatment, and evaluation of the initial urinary tract infection in febrile infants and young children. American Academy of Pediatrics. Committee on Quality Improvement. Subcommittee on Urinary Tract Infection. Pediatrics 103:843-852

Q5: Results UF-5000 BACT and WBC count and Figure 1. You mentioned in caption to fig.1 that fungi are excluded. Why is that? Are fungi defined as culture negative or positive? You need to define this or explain the rationale about excluding them. Fungi is a non-bacterial UTI.

Q6: You identified 513 samples with significant growth. I think you should use the same term in the manuscript for culture positive. So may change the sentence to 513 culture positive samples.

Q7: Diagnostic performance of UF-5000. Again rationale for excluding samples with fungi.

Reviewer #4: This study investigated the performance of Sysmex UF-5000 analyzer parameters for screening urinary tract infection (UTI) in children. I believe the manuscript should be rejected due to the following reasons:

1、 The manuscript is poorly organized and its readability needs to be extensively improved.

2、 The language quality does not meet the publication criteria of PLoS ONE.

3、 The authors should provide more details about the inclusion and exclusion criteria for the patient enrollment.

4、 I suggest the authors use a flowchart to depict participant selection.

5、 How to handle duplicated participants?

6、 The type of data collection (retrospective or prospective) should be reported.

7、 The sample sizes of UTI and non-UTI were not reported.

8、 In table 2, the rate of missed diagnosis should be reported.

9、 The novelty and strength of this study were not well discussed in the discussion section.

Minor comments

1、The upper limit of AUC’s confidence interval should not exceed 1.

2、The 95% confidence intervals for sensitivity and specificity should be reported

Reviewer #5: The authors report on the performance of the Sysmex system for diagnosing urinary tract infections among children from one hospital in China. The diagnostic method is valid and can reduce time to result and laboratory workload by reducing the number of unnecessary urine cultures. To better understand and interpret the findings, the authors should add more detail on the setting, study design and patient population. Further comments below.

Introduction

- Improper collection is more likely to lead to contamination and hence false positive results

- Comment on the use of dipstick test for UTIs in children (given that it also provides rapid results and is very cheap)

- When introducing Sysmex, please add the manufacturer and location

- Explain to the reader by which mechanism can the Sysmex differentiate between Gram positive and Gram negative organisms

Methods

- “If there is sterile growth” please rephrase to no growth or similar

- Describe what symptoms were considered and how the data were collected

- Add some information about where the study was conducted including some background about the setting, on the study design, prospective or retrospective, and on the approach for enrolling patients, alternatively whether data were obtained from medical records. Also include some data on the patient population to add with the interpretation of findings (enterococci were relatively common, Candida is usually considered a contaminant – were a considerable proportion of patients immunosuppressed)

- Data analysis: state in the text for which two parameters Sensitivity and specificity were computed

Results

- Add percentages in the first paragraph

- No need to list the SE for the ROC curve AUC

- Explain in the methods how the “best” cut-off was selected for calculating Sensitivity and specificity

- Would a combination of using WBC by Sysmex and Bacterial count improve test performance.

- “… based on the gold standard of midstream urine cultures, the total compliance rate of UF-5000 for distinguishing UTIs is…” Please note that when considering your definitions in the methods section, not all growth represents a UTI (some may be asymptomatic bacteriuria). If necessary correct for clarity. Also please clarify what is meant by “compliance rate”

- Explain what “unclassified flags” are

Discussion

- The yield of urine cultures is quite low. Comment in more depth on possible reasons (in adults yield can be higher than 40% reaching 70-80%). Similarly the prevalence of Gram-negatives was relatively low (one would expect that more than 60% of samples would have Gram negative growth) – comment on possible reasons.

- Replace Enterobacteriaceae with Enterobacterales

- When setting cut-off values for sensitivity and specificity consider that these can also be population-dependent as well as depend on the strategy the authors decide to use for setting the cut-off (better sensitivity vs better specificity vs Youden). These might explain the slight differences between studies and there might not be a “best” cut-off in this scenario (results apply to this specific study, as you discuss in the next sentence).

- “We recommend a BACT count cut-off of 7.8/uL for screening out patients with negative cultures, especially if they have no clinical symptoms of UTI.” – should these patients be tested at all if asymptomatic?

- Include the limitations before the conclusion

Tables and Figures

- Table 1: replace ratio with proportion, italicise species names, list in the footnote what other Gram positives and Gram negatives were isolated, also for other Fungus (other fungi); Proteus names is misspelled, consider that Candida can be a contaminant

- Table 2: add space between number and parantheses. There is typo for specimens in the first cell also for “total”

6. PLOS authors have the option to publish the peer review history of their article (what does this mean?). If published, this will include your full peer review and any attached files.

Reviewer #1: No

Reviewer #2: No

Reviewer #3: No

Reviewer #4: No

Reviewer #5: **Yes: **Ioana Diana Olaru

---

## [Author Response · Author response to Decision Letter 0]

6 May 2024

Dear Editor and Reviewers,

We have provided a detailed response to this section in “Response to Reviewers”.The revised portions are mark by “Track Change” in red font throughout the revised manuscript. 

Reviewers' comments:

Reviewer #1

1. Conclusions: I like that the authors write Gram usually with capital letter, because it is the name of a person. In Conclusion it should also be written with capital letter.

Reply: We greatly appreciate the reviewer’s comments. Following the reviewer’s suggestion, in the conclusion, the word “gram “was replaced with “Gram”.

2. Urine culture and identification of microbes. “If there was sterile growth”. Although the authors used 10 µl for urine culture, also in case of no growth, one cannot say it is sterile. Therefore, better: “If there was no growth”.

Reply: We greatly appreciate the reviewer’s comments. Following the reviewer’s suggestion, we use " If there was no growth " instead of " If there was sterile growth " in the revised manuscript.

3. Table 2. Typing error: Tocal? Maybe should read Total. It was used 3 times.

Otherwise I have no further comments.

Reply: Thank you for pointing out these errors. We have made the necessary corrections and conducted a thorough review of the spelling in the document.

Reviewer #2

1. Abstract (methods): “We analyzed 4445 patient information of children….” This sentence should be rewritten in a better and more informative way.

Reply: We sincerely appreciate the reviewer's feedback and have revised this section based on the suggestions provided. The revised portions are indicated by 'Track Changes' in red font in the updated version of our manuscript.

2. Ethics statement: “And began accessing data on December 29, 2023. The author had access to information that could identify individual participants during data collection…” This part should be rewritten in a better English.

Reply: We sincerely appreciate the reviewer's feedback and have revised this section based on the suggestions provided. The revised sections are highlighted with 'Track Changes' in red font in the revised version of our manuscript. 

3. Results/Microbial species: “C. tropical” should be spelled out to “Candida tropicalis”

Reply: We greatly appreciate the reviewer’s comments. In accordance with the reviewer's recommendation, we have updated the terminology in our manuscript to use “Candida tropicalis” instead of “C. tropical”.

4. Table 1: “Proteus mirabili” should read “Proteus mirabilis”

Reply: We greatly appreciate the reviewer’s comments. Following the reviewer’s suggestion, we use " Proteus mirabilis " instead of " “Proteus mirabili " in the revised manuscript.

5. Results/Diagnostic performance of UF-5000: “Excluding samples from the fungus culture and the < 104 CFU/mL group, an ROC curve…” should read “…, a ROC curve…”

Reply: Thank you for pointing this out. We have corrected these grammar mistakes mentioned above and have carefully reviewed the language throughout the document. 6. The “limitations” should be mentioned before the conclusion.

Reply: We greatly appreciate the reviewer’s comments and have adjusted the order of these two parts as suggested.

Reviewer #3

1.Abstract, Results. The author mentions that unnecessary culture could be avoided in 28.1%. Are there a reason for presented the numbers with 3 digits? Why not say 28%. I hardly think the estimates is that precise?

Reply: We appreciate the reviewer's feedback on our work. In order to minimize false negatives, a higher sensitivity of 99% was applied as a condition. The unnecessary cultures percentage was calculated as True Negative divided by the Total Number, resulting in 0.280839231547 (1111/3956). The information is presented in Table 2, where we have rounded the numbers to one decimal place to highlight subtle differences in the data. This approach was particularly important for metrics like negative predictive value (NPV) %. Therefore, we ensured consistency by consistently using one decimal place throughout the entire analysis process.

2.Abstract, Conclusion. Cultures screened with UF-5000 showed superior results in identifying…superior in comparison to what? I don’t think you mean compared to urine culture? Or do you mean compared to previous flow cytometry models like UF-1000 or compared to Urine stricks? The UF-5000 may be faster than culture but not better. It may also be better that other fast screening methods.

Reply: The reviewer's feedback is greatly appreciated. Our research found that cultures screened with UF-5000 demonstrated superior results in identifying negative cultures. Specifically, the consistency between UF-5000 BACT-Info flags indicating 'Unclassified' and negative urine cultures was 97.9% (3370 / 3443) Table 3). Moreover, Cultures screened with UF-5000 showed superior results in identifying significant bacterial growth, Specifically, For samples with ≥105 CFU/mL bacteria, “Gram-negative?” UF-5000 BACT-Info flags showed a better concordance of 74.1% (60/81) compared to 104-105 CFU/mL bacteria, with 57.5% (142/257) (highlighted in green in Table 3).Similarly, 53.4% (16/30) of urine cultures with growth ≥105CFU/ml of GP bacteria showed higher concordance than urine cultures with 104–105CFU/ml 34.8% (48/138) (highlighted in blue in Table 3). This sentence is indeed ambiguous, we appreciate the feedback provided and have revised the expression in the section accordingly, the revised portions are mark by “Track Change” in red font in the updated version of our manuscript.

3. Introduction/discussion. The authors mention that rapid screening test for UTI 

is needed for children. But there is nothing mentioned about urine dipsticks that are ruinously used in many settings for rapid screening, including in children >2 years of age. Is that because the dipsticks are not used at this children’s’ hospital? It may be worth mentioning that there are a lot of rapid screen test on the marked, but they do show inadequate performance compared to culture.

Reply: Our hospital predominantly utilizes the UC-3500 urine dry chemical analyzer for testing with dipsticks. This study focuses on exploring the new parameter of bacterial information provided by the UF-5000, and reduce unnecessary cultures based on bacterial count. We appreciate the reviewer for providing valuable feedback, which serves as a helpful reminder for our future research endeavors. In our next investigation, we plan to delve deeper into the joint testing of UC-3500 and UF-5000. As pointed out by the reviewer, the urine dipstick serves as a rapid screening test, and we have incorporated comments on its application for UTIs in the introduction of our revised manuscript. 

I concur with the reviewer's observation that the UF-5000 may be faster than culture but not better. Urine culture remains the gold standard for diagnosing UTIs。 and these contents are also reflected in the discussion.

4.Urine Culture: Inoculation involved using a calibrated loop to apply 10 µL culture. It is common to use a 1-μL inoculating loop, so can you elaborate on the choice of the 10 µL loop? It may be a good point to include in your discussion as many of the studies you compare uses 1 µL loop cultures and therefore you may not compare to the same culture standard.

Reply: We greatly appreciate the reviewer’s comments. Based on the health industry standard WS/T489-2016 of the People's Republic of China regarding 'Laboratory Diagnosis of Urinary Tract Infections', section 8.4.1 on Selection of Culture Types and section 10 on Interpretation of Results, it is recommended to use either a 1μL or 10μL inoculating loop. However, our hospital currently does not have the 1μL loop and therefore we typically use the 10µL loop. It is worth noting that while using the 10µL loop does not impact the judgment criteria. Furthermore, the 10μL loop can be effectively reused after proper sterilization with an alcohol lamp, offering a cost-saving measure. Please refer to the industry standard guidelines for more detailed information.

8.4.1 Selection of Cultivation Types

The clean midstream urine culture method involved inoculating 1μL or 10μL onto blood agar plates, MacConkey or Chinese blue agar plates. All plates were incubated for 18-24h at 37 °C in 5% CO2. If there was bacterial growth, the colonies were counted. If there was no growth, cultivation continued for 48h.

10 Result interpretation

10.1 Summary: With a 1μL inoculum, the count results in plate colony count×103 CFU/mL. With a 10μL inoculum, the count results in plate colony count×102CFU/mL. Clinical physicians are advised to analyze the clinical significance of urine culture results in conjunction with urine routine results.

10.2 General explanation: Following quantitative cultivation of clean midstream urine, a single bacterial colony count ≥ 105CFU/mL may indicate infection, while <104CFU/mL may be contamination. The range of 104CFU/mL to 105CFU/mL requires evaluation based on the patient's clinical manifestations. These parameters are usually sufficient for accurate diagnosis of most cases of pyelonephritis and cystitis.

 In conclusion, while various specifications of inoculating loop are employed, the judgment criteria remain consistent post-conversion.

5. Urine culture. BACT counts ≥ 104CFU/mL should be changed to BACT counts ≥ 104 CFU/mL. Do the sentence BACT counts ≥ 104CFU/mL (when symptomatic) where considered positive, mean that samples with ≥ 104CFU/mL where excluded if the patient did not have any symptoms? I don’t think you can exclude culture positive samples from the dataset, at least not without mentioning this an exclusion criteria. Also how where ≥ 103CFU/mL classified? As negative? You may stick to the recommended cut-off urine cultures in children at ≥ 104 CFU/mL.: 1999. Practice parameter: the diagnosis, treatment, and evaluation of the initial urinary tract infection in febrile infants and young children. American Academy of Pediatrics. Committee on Quality Improvement. Subcommittee on Urinary Tract Infection. Pediatrics 103:843-852

Reply: Culture positive samples were not excluded from the dataset, and all children with BACT > 104 CFU/mL were included in the research. We acknowledge the reviewer's feedback and have made the required revisions, as well as cited relevant articles in our manuscript. The revised sections are highlighted with 'Track Changes' in red font on pages 5-6 of the original text. Following industry standards where ≥ 103 CFU/mL is considered contamination, data with BACT < 104 CFU/mL were not included in the results of the second to fourth parts.

6. Results UF-5000 BACT and WBC count and Figure 1. You mentioned in caption to fig.1 that fungi are excluded. Why is that? Are fungi defined as culture negative or positive? You need to define this or explain the rationale about excluding them. Fungi is a non-bacterial UTI.

Reply: We appreciate the valuable feedback provided by the reviewer. It is important to note that fungi are considered a non-bacterial UTI, with the corresponding parameter in the Sysmex UF-5000 being YSL counts. While fungi are typically identified through culture positivity, our research parameters, specifically BACT counts and BACT-Info, primarily focus on bacterial UTIs. Additionally, the percentage of fungal samples in our study was relatively low at 1.2% (54 out of 4445 samples). Therefore, we made the decision to exclude fungal data in order to maintain the scientific integrity of our analysis. At present, we have not conducted a correlation study between fungal UTIs and YSL counts parameter due to the limited volume of available data. In order to express our research content more clearly, we have made modifications to the title：Enhancing clinical decision-making: Sysmex UF-5000 as a screening tool for bacterial urinary tract infections in children.

7. You identified 513 samples with significant growth. I think you should use the same term in the manuscript for culture positive. So may change the sentence to 513 culture positive samples.

Reply: We greatly appreciate the reviewer’s comments. Following the reviewer’s suggestion, we use " 513 culture positive samples " instead of " 513 samples with significant growth. " in the revised manuscript.

8. Diagnostic performance of UF-5000. Again rationale for excluding samples with fungi.

Reply: We appreciate the reviewer's comments and have incorporated corresponding explanations in question 6.

Reviewer #4

1.The manuscript is poorly organized and its readability needs to be extensively improved.

Reply: Thank you for your feedback. We have thoroughly revised the manuscript based on the reviewer's comments. We believe that our response and revisions address the concerns raised and enhance the quality of the publication. We hope that the revised version meets the standards for publication. The revised sections are mark by “Track Change” in red font throughout the revised manuscript. 

2.The language quality does not meet the publication criteria of PLoS ONE.

Reply: Thank you for your feedback. we have employing a professional scientific editing service to help us language editing.

3.The authors should provide more details about the inclusion and exclusion criteria for the patient enrollment.

Reply: This retrospective study utilized data extracted from medical records. Urine culture was employed as the reference standard for the research. To assess the performance of Sysmex UF-5000, it was imperative to ensure that the detection outcomes from both methods were derived from the same urine sample for comparability. Initially, data was gathered from pediatric patients who underwent urine sediment and urine culture tests on the same day. Subsequently, only the initial test results for each patient were retained based on their ID. Patients who underwent examinations on the same day but at different times, determined by the specific time of collection, were then excluded from the analysis. Finally, 4445 childrens were included in the study. 

4.I suggest the authors use a flowchart to depict participant selection.

Reply: We greatly appreciate the reviewer’s comments. Based on the reviewer's suggestion, we have incorporated a flowchart in the supporting information to describe the selection of participants. 

5.How to handle duplicated participants?

Reply: we will only retain the first test data for each patient based on their unique patient ID. 

6.The type of data collection (retrospective or prospective) should be reported.

Reply: As a retrospective study, this detail has been included in the methodology section.

7.The sample sizes of UTI and non-UTI were not reported.

Reply: Thank you for your feedback. Based on the health industry standard WS/T489-2016 of the People's Republic of China regarding 'Laboratory Diagnosis of Urinary Tract Infections' and reference[14]：BACT counts ≥ 104 CFU/mL were considered culture positive, BACT counts < 104CFU/mL raised suspicion of contamination [14]. The urine was classified as contaminated if three or more colonies without a predominant organism were observed, representing the diagnostic standard for pediatric patients in the author's institution. We divided 4445 children into the following groups(In Response to Reviewers).

This study aims to validate the performance and intended use of the novel UF-5000 fluorescence flow cytometer. The findings will be compared to culture results to evaluate the accuracy of the UF-5000 in discriminating between gram-positive and gram-negative bacteria. Therefore, research results were not grouped according to UTI and non-UTI categories, but instead, utilizing grouped data for analysis can provide a clearer demonstration of our research results.

8.In table 2, the rate of missed diagnosis should be reported.

Reply: According to the reviewer's feedback, we have added the missed diagnosis rate, which is the false negative rate in Table 2.

9.The novelty and strength of this study were not well discussed in the discussion section.

Reply: We greatly appreciate the reviewer’s comments. The Sysmex UF-5000, a third-generation fully automated urine flow cytometer, a new function is its ability to distinguish gram-positive and gram-negative bacteria in urine samples. The majority of previous studies have focused on adult samples. To this end, we a

---

## [Editor Report · Decision Letter 1]

9 May 2024

Enhancing clinical decision-making: Sysmex UF-5000 as a screening tool for bacterial urinary tract infections in children

PONE-D-24-04363R1

Dear Dr. Xin Lv,

We’re pleased to inform you that your manuscript has been judged scientifically suitable for publication and will be formally accepted for publication once it meets all outstanding technical requirements.

Kind regards,

Eleonora Nicolai, PhD

Academic Editor

PLOS ONE
---

## [Editor Report · Acceptance letter]

3 Jun 2024

PONE-D-24-04363R1 

PLOS ONE

Dear Dr. Lv, 

I'm pleased to inform you that your manuscript has been deemed suitable for publication in PLOS ONE. Congratulations! Your manuscript is now being handed over to our production team.

Kind regards, 

on behalf of

Dr. Eleonora Nicolai 

Academic Editor

PLOS ONE